# Elevated Expression of the Immune Checkpoint Ligand CD276 (B7-H3) in Urothelial Carcinoma Cell Lines Correlates Negatively with the Cell Proliferation

**DOI:** 10.3390/ijms23094969

**Published:** 2022-04-29

**Authors:** Niklas Harland, Florian B. Maurer, Tanja Abruzzese, Cornelia Bock, Ivonne A. Montes-Mojarro, Falko Fend, Wilhelm K. Aicher, Arnulf Stenzl, Bastian Amend

**Affiliations:** 1Department of Urology, University of Tuebingen Hospital, Hoppe-Seyler-Str. 3, 72076 Tuebingen, Germany; Niklas.Harland@med.uni-tuebingen.de (N.H.); urologie@med.uni-tuebingen.de (A.S.); 2Center for Medical Research, University of Tuebingen Hospital, Waldhoernlestr. 22, 72072 Tuebingen, Germany; f.maurer1@web.de (F.B.M.); Tanja.Abruzzese@med.uni-tuebingen.de (T.A.); conny.bock@med.uni-tuebingen.de (C.B.); aicher@uni-tuebingen.de (W.K.A.); 3Institute for Pathology, Eberhard-Karls-University, 72072 Tuebingen, Germany; Ivonne.Montes@med.uni-tuebingen.de (I.A.M.-M.); Falko.Fend@med.uni-tuebingen.de (F.F.)

**Keywords:** urothelial carcinoma, immune checkpoint antigen, CD276, bladder cancer stem cell

## Abstract

The cell surface molecule CD276 (B7-H3) is an immune checkpoint antigen. The elevated expression of CD276 on tumors contributes to the suppression of anti-tumor T-cell responses and correlates with poor prognosis. Methods: The expression of CD276 was explored in vitro on eight urothelial carcinoma cell lines (UM-UC) in comparison to eight normal urothelial cells (NUCs) by RT-qPCR, Western blotting, and flow cytometry. Cell proliferation was enumerated over consecutive passages. The expression of cancer stem cell markers CD24 and CD44, cytokeratins, and vimentin was investigated by immunofluorescence. The expression of CD276 in bladder tumor samples and metastases was explored by immunohistochemistry. Results: Expression of CD276 on cell surfaces was elevated on UM-UCs when compared to NUCs. In UM-UCs, CD276 transcripts correlated moderately positive with CD276 protein expression (ρ = 0.660) and strongly positive with CD276 surface-expression (ρ = 0.810). CD276 mRNA expression (ρ = −0.475) and CD276 protein expression (ρ = −0.417) had a significant negative correlation with proliferation, while a significant correlation between proliferation and cell surface expression was not observed in UM-UCs. Conclusion: The expression of CD276 on UM-UC bladder tumor cell surfaces is elevated. Slow proliferating UM-UC cells express more CD276 mRNA and protein than fast proliferating cells. In patients, slow proliferating CD276^high^ tumor (stem) cells may evade immune surveillance. However, cancer therapy targeting CD276 may be effective in the treatment of slow proliferating tumor cells.

## 1. Introduction

Urothelial cell carcinoma (UCC) is the most common malignancy of the urinary system [1]. It is associated with inherited factors such as genetic disposition polymorphism in some loci, as well as with acquired factors including the mutation of some oncogenes or tumor suppressors [2]. The exposition of individuals to environmental risk factors such as aromatic amines increases the risk of developing a UCC [3]. The majority of UCCs are superficial non-muscle-invasive urothelial tumors. These tumors frequently show mutations of fibroblast growth factor receptor 3 (FGFR3) [4]. Such vesical tumors can be resected. However, they are known for frequent recurrence. Superficial tumors do not commonly develop in muscle infiltrating tumors, and metastases are comparably rare. Still, about 20% of UCC patients present with muscle-invasive bladder cancer and with metastases [5]. As some UCCs remain undetected until the patient develops an aggressive tumor, they are not treated in the early cancer stages. Therefore the 5-year survival rate of bladder cancer patients has not improved in recent decades despite manifold and intensive research [1,6].

In bladder cancer tissue elevated expression of immune checkpoint antigens such as programmed cell death protein-1 (PD-1; =CD279), its ligand, PD-L1 (=CD274), anti-cytotoxic T-lymphocyte associated antigen-4 (CTLA-4, =CD152), and B7-H3 (=CD276) was observed [7,8,9]. These factors modulate T-cell mediated immune responses. The elevated expression of CD47 on tumor cells suppresses the activation of macrophages [10]. The overexpression of these factors may prevent anti-tumor responses by the immune system [11]. At the same time, overexpression of these factors in tumors may open new avenues for immunotherapy using antibodies targeting these checkpoint antigens [12,13,14]. However, immune checkpoint antigens are not only protein targets for cytotoxic antibody therapy [14]. They are also involved in intercellular and intracellular signaling events [15]. Treating the microenvironment of a tumor and thus modulating tumor cell communications may therefore be an even more promising strategy when compared to targeting immune checkpoint antigens on tumor cell surfaces only [16]. Tumor cell lines expressing immune checkpoint antigens are valuable tools to explore such strategies in in vitro models.

The elevated expression of CD276 was reported on a variety of cancer cells, including bladder cancer cells [9,17,18]. It was considered a promotor of metastases [19], paving the way for muscle-invasive bladder cancer [20,21], and associated with poor prognosis [22,23]. The elevated expression of CD276 was also reported on cancer stem cells, promoting cancer cell proliferation and expansion of the cancer stem cell pool [24,25]. We therefore investigated the expression of CD276 on eight urothelial carcinoma cell lines of the University of Michigan Urothelial Carcinoma (UM-UC) Cell Repository [26] on the transcript, total protein, and cell surface expression levels in correlation to the in vitro growth patterns and proliferation rates.

## 2. Results

### 2.1. Expansion of Normal Urothelial Cells and Urothelial Cancer Cell Lines

Normal human urothelial cells (NUCs) and human bladder cancer cell lines of the UM-UC series were expanded in vitro. The cultures presented with quite distinct features. Upon seeding, NUCs and UM-UC-13 attached as individual cells and proliferated to yield confluent cultures (Figure 1A,F). In contrast, dense and compact clusters were seen with lines UM-UC-5, -6, -14, and -16 (Figure 1B,C,G,I), while lines UM-UC-9, -10, and -15 generated wider clusters (Figure 1D,E,H). NUCs proliferated with a mean duplication rate of 0.193 cell cycles per 24 h. All UM-UC lines proliferated at 3- to 5-fold higher proliferation rates (Figure 2). A correlation between growth patterns or appearance of cells (Figure 1) and proliferation rates (Figure 2) was not observed.

### 2.2. Expression of CD276 by Normal Urothelial Cells and Urothelial Cancer Cell Lines

The expression of immune checkpoint antigen CD276 by NUCs and UM-UC cells was investigated on the level of transcripts, total proteins, and on cell surfaces (Figure 3). In contrast to bladder cancer cell lines HT1197, TCCsup, RT4, and 5637 investigated recently [9] a significant overexpression of CD276 transcripts was not observed in the cells of the UM-UC bladder cancer cell lines (Figure 3A). UM-UC-13 expressed significantly less CD276 transcripts when compared to NUCs (*t*-test: *p* = 0.015). The normal distribution of CD276 transcripts was tested using Shapiro–Wilk test with *p* = 0.37 confirming normal distribution (skewness 0.29, kurtosis −0.866). The expression of total CD276 protein in NUCs and UM-UC lines followed at large the transcript amounts. (Figure 3B,C). The amount of C276 on NUC and UM-UC cell surfaces was determined by flow cytometry (Figure 3D). While UM-UC13 expressed rather low levels of CD276 on cell surfaces (MFI 1746 ± 1072; *n* = 5), UM-UC-10 expressed significantly more CD276 (MFI 6857 ± 2433; *n* = 5, *p* < 0.0041). Overall, the UM-UC lines investigated presented more CD276 on cell surfaces (MIF 9335 ± 3330; *n* = 8) than NUCs (Figure 3E,F). In a separate study, NUCs expressed low CD276 levels as well (MFI 1043 ± 97; *n* = 6, not shown). Taken together, the highest cell surface expression was observed on UM-UC-15 cells (Figure 3D,E). At the same time, UC15 expressed only moderate levels of total protein when compared to NUCs (Figure 3C). This indicated that the CD276 protein expression levels did not correlate tightly with its staining intensities on cell surfaces in UM-UC-15. Moreover, it suggested that translocation of the CD276 protein to the cell surface was possibly regulated independently from transcription and translation in some UM-UC cell lines.

### 2.3. Analysis of Correlation between Cell Proliferation and CD276 Expression

The correlations between in vitro cell growth of UM-UC cell lines and expression of immune check-point antigen CD276 were computed and the Spearman’s rank correlation coefficient ρ is presented for the corresponding data sets. (Figure 4). The cell duplication rates revealed a significant negative correlation with CD276 transcript expression (ρ = −0.475, *p* < 0.05; Figure 4A) and total protein expression (ρ = −0.417, *p* < 0.05; Figure 4B). However, the cell proliferation did not correlate significantly with CD276 on cell surfaces (ρ = 0.240, n.s.; Figure 4C). CD276 transcripts correlated significantly with CD276 total protein (ρ = 0.660, *p* < 0.01; Figure 4D) and cell surface presentation (ρ = 0.81, *p* < 0.05; Figure 4E). However, total protein expression did not correlate significantly with CD276 expression on cell surfaces (ρ = 0.619, n.s.; Figure 4F). We conclude that elevated expression of CD276 transcripts and total protein is not associated with a fast proliferative phenotype in UM-UC cells.

### 2.4. Expression of Tumor Stem Cells Markers

The elevated expression of CD276 was considered a bladder cancer stem cell marker [17]. Others considered the expression of the aldehyde dehydrogenase (ALDH1) paralog A1 [27], or the expression of CD24 and CD44 as bladder cancer stem cell markers [28,29]. We therefore investigated if the expression of CD276 correlated with the expression of ALDH1-A1, CD24, and CD44 on the UM-UC cells (Figure 5). The highest expression of ALDH1-A1 transcripts was recorded in UM-UC-15, while UM-UC-6, -9, -10, and -16 expressed low levels (Figure 5A). On average, the expression of paralogs ALDH1-A2 and -A3 was 2 logarithms lower than the expression of ALDH1-A1 (not shown). Only in UM-UC-6 noteworthy ALDH1-A2 and -A3 transcript levels were recorded (not shown), while this line expressed the lowest levels of ALDH1-A1 (Figure 5A). The expression of CD24 was highest in UM-UC10, UM-UC-15, and UM-UC-5 (Figure 5B), while CD44 was expressed yet a different pattern, and remarkable steady-state transcript levels were recorded only in UM-UC-5 cells (Figure 5C). A significant correlation between tumor markers ALDH1-A1, -A2, and -A3, CD24, CD44 on one hand, and CD276, on the other hand, was not computed. We conclude that elevated cell proliferation did not correlate with elevated expression of the bladder cancer stem cell markers investigated in the UM-UC lines studies here.

### 2.5. Determination of the Cell Lineage of UM-UC-13 Bladder Cancer Cells

The UM-UC-13 cells were derived from a metastasis in a lymph node of a bladder cancer patient. In contrast, all other UM-UC lines included in this study were isolated from primary bladder cancer tissue samples [30]. We therefore investigated the expression of cytokeratins (CKs) as a urothelial marker in comparison to the mesenchymal lineage marker vimentin on UM-UC-13 in comparison to UM-UC-10 and NUCs. On NUCs, a prominent expression of CKs, detectable by antibody cocktail AE1/AE3, was observed (Figure 6A), and some cells expressed vimentin (Figure 6B). This confirmed that early passage NUCs were a blend of urothelial cells complemented by some mesenchymal cells. The prominent expression AE1/AE3 CKs was observed on UM-UC-10, but the elevated expression of vimentin was not detected by immunocytochemistry (ICC; Figure 6E,F). This corroborated their urothelial phenotype. The UM-UC-13 cells detached during the fixation and ICC staining procedures in 4 consecutive attempts (Figure 6I–L). This difference in cell attachment correlated with the distinct growth patterns on culture vessels observed for UM-UC-13 (Figure 1F). Therefore, the expression of CKs, CD276, and vimentin was investigated on UM-UC-13 by a different protocol employing immunofluorescence (Figure 7). The expression of CK8/18 and CK5 (Figure 7A–C) was recorded, and AE1/AE3 staining was detectable in UM-UC-13 (Figure 7E,F). The low expression of CD276 was recorded on UM-UC-13 by extended exposure (2.7 s, Figure 7G,H), but vimentin was not detected at all (Figure 7I, J). This confirmed that UM-UC-13 was a urothelial cell and at the same time the low CD276 expression patterns were observed on transcript and protein levels and on the cell surfaces (Figure 3).

### 2.6. Detection of CD276 in Lymph Node Metastases of Bladder Cancer Patients

The expression of CD276 was considered a promotor of metastases [19], and the expression of CD276 was also reported on cancer stem cells [24,25]. We therefore investigated the expression of CD276 as well as CD24 and CD44 in lymph node metastases of bladder cancer patients (Figure 8A–F) in comparison to bladder cancer samples (Figure 8G–L). Prominent staining with AE1/AE3 confirmed infiltration of urothelial cells in all lymph nodes investigated (Figure 8A). The expression of mesenchymal marker vimentin was also observed in all samples (Figure 8B). The expression of tumor stem cell marker CD24 was not detected in any of the 5 lymph node samples investigated (Figure 8C), but CD44 positive cells were observed in each sample (Figure 8D). The prominent expression of C276 was noted in the infiltrates of all samples (Figure 8E). In tumor samples from bladder tissue (*n* = 5), the expression of AE1/AE3 antigens (Figure 8G), and vimentin (Figure H) were found in all of the investigated samples. The expression of the UCC stem cell marker CD24 was observed in samples from 3/5 tissues (Figure 8I), while the CD44 was detected only on a few cells (Figure 8J). The expression of CD276 was detected in all of the investigated samples (Figure 8K). We conclude that not all bladder cancer tissue samples contained considerable numbers of CD24-positive cells. The CD24-expressing tumor stem cells did not home to the metastases in the lymph nodes, whereas CD276-expressing cells were found in the metastases.

To investigate if CD276 positive cells in lymph nodes are in a proliferative stage, the co-expression of CD276 and Ki67 was explored in lymph node infiltrating metastases (Figure 9). The analyses of consecutive samples from 5 donors documented a strong co-expression of these antigens. These results suggested that the lymph node-derived urothelial carcinoma cell line UM-UC-13 had not conserved the Ki67^hi^CD276^hi^ phenotype observed in metastases from bladder cancer patients investigated in this study (Figure 2, Figure 3 and Figure 9).

## 3. Discussion

In this study, we explored the expression levels of CD276 and other bladder cancer stem cell markers including ALDH1, CD24, and CD44 as a function of cellular proliferation on UM-UC bladder cancer cells and NUCs. In addition, paraffin sections from bladder cancer tissue and metastases in lymph nodes were investigated for expression of CD276, CD24, and CD44. The duplication rates of the UM-UC bladder cancer lines investigated were 3- to 5-fold higher than the proliferation rates of normal urothelial cells. This is not surprising, as these UM-UC lines have undergone a selection of fast proliferating cells by long-term in vitro culture. However, duplication rates correlated negatively with CD276 transcript (ρ = –0.475) and total protein (ρ = –0.417) expression. This indicated that the elevated expression of CD276 was not a selection marker in the UM-UC lines investigated nor enhanced during extended in vitro culture. It rather confirmed the elevated expression of this immune checkpoint antigen in bladder cancer-derived cells [9]. A significant correlation between cell surface presentation of CD276 and proliferation rates was not seen (ρ = 0.240). Slow proliferating UM-UCs expressed significantly less CD276 mRNA and protein. The CD276 transcript rates correlated significantly with the protein expression (ρ = 0.660 **) and cell surface presentation (ρ = 0.810 *). The protein expression only corresponded to the cell surface presentation (ρ = 0.619); however, it did not correlate significantly. This indicated that the shedding of CD276 from UM-UCs and generation of soluble CD276 was a not major mechanism to explain differences in CD276 expression in UM-UC cells as reported for other cells [31]. Moreover, the analyses of UM-UC-15 provided evidence of prominent CD276 cell surface presentation despite low transcript amounts, which are translated in UM-UC-15 in CD276 protein expression level below NUCs. This suggested an efficient transport of CD276 to the cell surface in UM-UC-15. Many UM-UCs expressed the CD276 protein in the range of NUCs. Only minor differences in CD276 total protein expression were recorded among the UM-UC cells. By contrast, a trend of elevated CD276 cell surface expression was evident in most UM-UC cells. The same applies to the proliferation rates of NUCs in comparison to UM-UC cells. The overall low protein expression as well as the rather moderate differences in total CD276 protein expression among UM-UC cells may contribute to the fact that the total CD276 protein expression did not yield significant correlations to the cell proliferation or cell surface staining intensities.

However, knowledge of the cellular distribution of CD276 in cancer cells is of interest as a recent study provided evidence that cytoplasmic expression of CD276 was associated with shorter disease-specific survival of cancer patients [32]. This would be in line with the function of CD276 as an immuno-modulatory antigen on cell surfaces [33]. Most studies investigating the role of immune checkpoint antigens on cancer cells focused on the plasma membrane expression and immune-modulatory effects [33,34,35,36,37,38,39,40,41]. Silencing CD276 in human lung cancer cells inhibited cellular invasion, mitosis, and migration by integrin-dependent mechanisms [42]. As a member of the CD28/CD80 integrin-like receptor-ligand family, it is considered a ligand [18,43]. But, at least in humans, knowledge of a natural CD276 receptor is sparse. Therefore, little is known if in human tumor cells an outside-in signaling through CD276 may contribute to the pathology of cells affected. In mice, the TREM-like transcript 2 expressed on T lymphocytes is a CD276 receptor modulating the T cell receptor signaling, thus limiting T-cell responses against tumor cells [44]. Again, little is known about ligand signaling in the CD276 expressing cell. In bladder cancer cells, CD276-mediated activation of PI3-Kinase-dependent pathways was observed [20]. Crispr-Cas9-mediated knocking down of CD276 in colorectal cancer cells increased Dickkopf-related protein 1, plasminogen activator-1, and urokinase plasminogen activator 1 [41]. STAT3-signaling by CD276 was reported [45]. CD276 shares intrinsic effects by regulating aerobic glycolysis and cell division [18,46,47]. These effects seem to be modulated by intracellular CD276. This is clear evidence for CD276-mediated cell responses independent of its cellular localization.

For the transport of CD276 to the cell surface, proteins must be encapsulated by coated vesicles, which shuttle cargo between the endoplasmatic reticulum and the Golgi apparatus. From Golgi, three classical export routes using secretory vesicles, a direct route, or endosomes may transport CD276 to the cell surface [48]. Additional pathways for targeting the plasma membrane independent from vesicle-mediated and Golgi-dependent routes were described as well [49]. We therefore conclude that in UM-UC-15 even moderate amounts of CD276 proteins are sufficient for a comparably high density of this antigen on the cells. A detailed study on mechanisms contributing to enhanced transportation of CD276 to the cell surfaces may provide new aspects of upregulation of this immune checkpoint antigen in bladder cancer. However, experiments along these lines must await additional research. CD276 expression is also modulated by non-coding micro RNAs (miRs), and 17 miRs were recently reported to interact with the CD276 encoding transcript [19]. However, none of the UM-UCs investigated here presented with low cell surface staining and particularly high transcript and/or protein expression levels at the same time. We therefore exclude this mode of CD276 regulation for the tumor lines investigated here [32]. The positive correlations between transcript and protein expression on one side and cell surface staining on the other side also exclude mechanisms of CD276 release as reported for instance for activated peripheral blood mononuclear cells [31]. This mechanism could generate CD276^low^ tumor cells which could become more accessible to anti-tumor immune responses. However, for its function as an immune-modulatory antigen, elevated cell surface expression of CD276 is relevant [18]. When analyzing bladder cancer tissue samples by immunohistochemistry or other antibody-dependent methods, low protein expression may therefore underscore the immunological potential of CD276. When planning for tumor therapy, by targeting tumor cells by anti-CD276 therapy, underscoring CD276 expression by immunohistochemistry should take the seeming discrepancies between total CD276 protein in the cell versus on the cell surface into account.

In recent studies, the expression of CD276 in bladder cancer tissue samples was investigated and a wide variety of CD276 staining intensities was found in both the pathologically benign-appearing samples from bladder cancer patients (score 0 to approx. 150) in comparison to pathologically define malignant areas (score 0–300) confirming previous studies [9,50]. This variety in CD276 expression was corroborated by this study when comparing for instance the normalized transcript amounts in lines UM-UC-13 and -16 versus UM-UC-9 or 10. UM-UC-13 and -16 transcribed even less CD276 encoding mRNA when compared to NUCs. In the total protein levels, the differences between NUCs and UM-UC lines seem less prominent, but on the cell surfaces CD276 expression levels above NUCs were recorded for all UM-UCs—with the exception of UM-UC-13. UM-UC-13, isolated from a metastasis in a lymph node from a bladder cancer patient [30], differs from the other UM-UC cells with respect to attachment and growth patterns in vitro as well. However, it maintained a urothelial phenotype and did not gain a prominent expression of the mesenchymal marker vimentin. Thus, the difference between UM-UC-13 and the other UM-UC lines is not associated with an epithelial-mesenchymal transition of bladder cancer cells [51,52].

The elevated expression of CD276 was considered a bladder cancer stem cell marker. We therefore investigated the expression levels of other BC stem cell markers in UM-UCs as well. Comparably high expressions of ALDH1-A1 and CD24 were recorded in UM-UC-15. However, CD44 was detected at comparably moderate levels in these cells. UM-UC-15 expressed CD276 transcript levels in the range of NUCs, and a simultaneous and significant upregulation of the four BC stem cell markers investigated in this study was not found. In the other UM-UC lines variable expression patterns of CD276, ALDH1-A1, CD24, and CD44 were observed, supporting the notion that evidence for a concurrent activation of stem cell marker expression is lacking in the UM-UC cells.

The knock-down of CD276 expression in melanoma cells decreased their metastatic potential. It therefore was considered a promotor of metastases [19,21,53]. Comparably, CD24 and CD44 were considered metastasis-associated molecules [54,55]. We therefore compared the expression of these antigens in lymph node metastases with the expression in bladder cancer tissue. The expression of CD24 was not observed in the metastases investigated, but CD44 and CD276 were found. In contrast, a rather strong expression of CD24 was found at least in some areas of bladder cancer samples, but only a few cells expressed CD44 at barely noticeable levels, while CD276 positive cells were found in all samples. The immunohistochemistry applied here does not qualify as precise methods in a strictly technical term. Therefore, the expression levels reported here should be considered descriptive, but not quantitative. However, the lack of CD24 expression in all metastases and little expression of CD44 in bladder cancer tissues may reflect the wide variety of phenotypes found in malignant cells contributing to the cancer etiology. We conclude that CD24^pos^ bladder cancer stem cells are not enriched in metastases of lymph nodes, but CD276 is found on both sides. The knock-down of CD276 by a lentiviral vector decreased cell proliferation and induced apoptosis in progenitor cells. The silencing of CD276 of such cells arrested mitosis in the G0/G1 stage of the cell cycle [56], while its overexpression promoted the G2/M phase transition of the cell cycle [45]. This suggested that UM-UCs seem to not utilize the CD276-dependent mechanisms involved in G0/G1 arrest, nor the promotion of G2/M phase transition [45,56]. We therefore investigated the co-expression of CD276 and the proliferation marker Ki67 in BC tissue samples. Using consecutive paraffin sections an exact co-expression of CD276 and Ki67 was recorded in metastases of bladder cancer. This is in line with other reports pointing towards the role of CD276 in tumor cell proliferation [20,45], but it contrasts with the results obtained with the UM-UC cells investigated here. We hypothesized that slow proliferating carcinoma cells may express CD276 at elevated levels, while in the metastases fast proliferating tumor cells are enriched. Taken together, we confirmed the overexpression of CD276 on cell surfaces of UM-UCs when compared to NUCs. At the same time, individual UM-UCs expressed quite different levels of CD276 mRNA and protein. The elevated expression of CD276 transcripts correlated well with the expression of CD276 on the total protein levels and cell surfaces. We concluded that in UM-UCs, the main regulatory pathways controlling the CD276 expression are associated with its promotor and transcription, while in other bladder cancer cells, the regulatory effects of micro-RNAs on CD276 protein expression were observed [19]. The pathways regulating CD276 cell surface presentation on UM-UCs and on other bladder cancer cells must await additional experiments.

## 4. Materials and Methods

### 4.1. Preparation and In Vitro Culture of Cells

Normal urothelial cells (NUCs; *n* = 8) were obtained from pathologically healthy-appearing sites of surgical samples from patients after informed and written consent. NUCs were prepared, expanded, and characterized as described recently [9]. Urothelial carcinoma cell lines UM-UC-5, -6, -9, -10, -13, -14, -15, and -16, originally from the University of Michigan Urothelial Carcinoma Cell Repository (supplied by Prof. P. Black, Vancouver Prostate Centre, University of British Columbia, Canada [57]), were seeded at an inoculation density of 5E05 per flask and cultured in MEM Eagle medium, enriched by 10% FBS, 1% non-essential amino acids, and antibiotics (all from Gibco Life Technologies, Carlsbad, CA, USA) [30]. At about 80% of confluency, cells were detached (Trypsin-EDTA), washed by PBS, counted with the aid of trypan blue staining and a hematocytometer, and split into several flasks to yield a starting density of 1E05 cells per flask. Cell proliferation was determined for each line over at least 3 consecutive passages. Contamination of the cultures by mycoplasma was excluded using a PCR-based technique following established protocols (MycoSPY; Biontex, Munich, Germany). The study was approved by the local Ethics Committee (file number 8041/2020/BO2).

### 4.2. Analysis of mRNA Transcripts

The quantification of gene expression on mRNA transcript levels was performed as described recently [9]. In brief, the cells were washed, and counted. Total RNA was extracted, DNA was degraded enzymatically, and total RNA was isolated using kits (RNeasy, Qiagen, Hilden, Germany). The yield and purity of total RNA were determined by UV spectroscopy (Nanodrop; Implen, Munch, Germany). Then, 1 μg of total RNA was reverse transcribed employing oligo-(dT) primers (Advantage RT-for-PCR kit TakaraBio, Saint-Germain-en-Laye, France). Quantitative PCR (LightCycler 480; Roche, Pensberg, Germany) of cDNA corresponding to the target genes—CD276, ALDH1-A1, -A2, and -A3, and CD24 and CD44—was performed as described recently [9]. The quantification of transcripts encoding GAPDH and PPIAγ served as controls. The amounts of target transcripts were normalized in each batch to the controls as described [58].

### 4.3. Analysis of CD276 Protein Expression

For extraction of total protein, cells were washed twice with PBS, detached by the aid of Accutase, aspirated, washed again, counted, and sedimented by centrifugation (200 G, 7 min, 4 °C). 5E05 cells were resuspended in 100 μL RIPA+ buffer (CC Pro; Oberdorla, Germany). Samples were homogenized by vigorous pipetting and stored at −80 °C. For further analyses, samples were thawed on wet ice and mixed by vortexing. To remove insoluble debris, samples were centrifuged (20,000 G, 20 min, 4 °C) and aliquots of the supernatant were transferred to fresh tubes and stored at −80 °C. The protein concentration in the precleared supernatant was determined by spectrophotometry as suggested by the supplier (DC Protein Assay kit II; Bio-Rad, Feldkirchen, Germany). For Western blotting, samples were mixed with Lämmli protein gel loading buffer to yield 50 μg of protein per lane and separated by SDS-PAGE as described [59]. The proteins were transferred to Nylon membranes by submarine electrophoresis. The membrane was washed three times with 0.1% (*v*/*v*) Tween20 in PBS (PBS-T) to remove SDS, blocked by 5% dry milk (*w*/*v*) in PBS-T (1 h, ambient temperature (AT), slow shaker), washed again in PBS-T and incubated overnight with anti-CD276 antibody (mo mAB IgG; abcam, Cambridge, UK, 1:600 in 5% dry milk (*w*/*v*) in PBS-T, 4 °C, slow shaker). The primary antibody solution was removed, the blot was washed (3× PBS-T, 20 min, AT), and reacted with HRP-labelled rb- anti-mo IgG (Dako-Agilent, Hamburg, Germany), 1:10,000 in PBS-T, 1 h, AT, slow shaker). The blot was washed 3× with PBS-T at AT [59]. To visualize the antibodies, the blot was reacted with the chemiluminescence substrate and recorded in a scanner as described by the supplier (Li-Cor, C-DiGit, Bad Homburg, Germany). Signal intensities were quantified by the proprietary software program (Image Studio Lite; Li-Cor). Then, the antibodies were stripped off, the membrane was blocked, and prepared for a second primary antibody incubation as described above. The quantification of β-actin (rb-anti-hu β-actin; abcam, 1:1000; followed by gt-anti-rb Ig, Dako, 1:2000) served as internal control and the expression levels of CD276 were normalized in each sample to the signal intensity recorded for β-actin.

### 4.4. Analysis of CD276 Presentation on Cell Surfaces

The expression of CD276 on cell surfaces was enumerated by flow cytometry (FC) as described recently [9]. In brief, cells were harvested by mild proteolysis (Accutase), washed, and sedimented in microtubes. 5E05 cells were resuspended in FC staining buffer (PFEA) enriched with 50 μL Gamunex (Grifols, Clayton, NC, USA) to block unspecific binding of antibodies to cell surfaces. Gamunex was washed off, and the cells were resuspended in PFEA buffer complemented with PE-labelled mo-anti-hu CD2756 IgG mAB (351004, BioLegend, San Diego, CA, USA 1:20). The samples were incubated at 4 °C in the dark for 1 h. The unbound antibody was removed by washing the cells twice, and CD276 staining was investigated by FC (LSR II, BD Bioscience, Franklin Lakes, NJ, USA), using FACS Diva (BD Bioscience) and FlowJo (Treestar, Ashland, OR, USA) software programs. Unstained cells and COMP-beads (BD Bioscience) served as controls to normalize staining intensities. Data are presented as mean fluorescence intensities (MFI) of stained cells (white histograms) vs. controls (gray histograms).

### 4.5. Immunocytochemistry and Immunohistochemistry

To detect the expression of cell surface proteins on cultured cells, immunocytochemistry (ICC) was performed. Cells were washed 3× from 2 min with PBS, fixed (4% paraformaldehyde, 10 min, AT), and washed twice again with PBS. The cells were permeabilized by saponin (1:50 in PBS, 10 min. AT, Sigma-Aldrich, Steinheim, Germany) and washed twice by PBS again. Unspecific peroxidase activity and antibodies binding were blocked as recommended (IHC ZytochemPlus HRP or AP polymer kit, abcam), and samples were washed again. Antibodies to cytokeratins detected by AE1/AE3 (MAB3412, 1:500 in PBS, Millipore) as epithelial and urothelial markers or anti-vimentin (550513, 1:100 in PBS, Becton-Dickinson) as mesenchymal antigen were added and incubated in a humidified box (37 °C, 2 h). The primary antibodies were washed off (3× for 5 min, PBS), and detected by the HRP or AP polymer reagent (IHC ZytochemPlus). Samples omitting the primary antibodies served as controls. The detection reagents were washed off (3× for 5 min, PBS) and the samples were counterstained by HE, covered (VectaMount, Vectorlabs, Toronto, Canada), and recorded by microscopy (Axiovert A1, Zeiss).

In addition, for UM-UC-13 cells immunofluorescence (IF) was employed. Antibodies AE1/AE3 (see above), anti-vimentin (see above), and anti-CD276 (ab226256, 1:100 in PBS, abcam) were added to the cells, incubated, and washed as described above. The binding of primary antibodies was detected by Cy-3 labeled gt-anti-mo affinity-purified antibodies (1:1000, Jackson Immuno Research, West Grove, PA, USA) or AlexaFluor 488-labeled gt-anti-rb antibodies (1:500, Jackson Immuno Research). The incubation of samples with mouse (clone R312; 1:500, abcam) or rabbit (clone EPR25A; 1:500, abcam) IgG1 antibodies served as isotype controls. Samples omitting the primary antibodies served as secondary antibody controls. Cell nuclei were labeled by DAPI. The staining of cells was visualized and recorded by fluorescence microscopy (Axiophot A1, Zeiss).

To determine the expression of cytokeratins, CD276, CD24, and CD44 on lymph node metastases from bladder cancer patients and on the primary tumor of the bladder, immunohistochemistry (IHC) was performed. Microsections (5 μM) were generated from paraffin-embedded samples from lymph node metastases of 5 bladder cancer patients and from 3 bladder cancer tissue samples (RM 2125RT, Leica, Wetzlar, Germany). The paraffin sections were rehydrated and incubated with primary antibodies to AE1/AE3, CD276, CD24 (ab202073 1:50, abcam) and CD44 (ab9524, 1:50, abcam) as described above. HE-staining was employed to visualize the histology of the tissue sections. The microsections were counterstained and recorded as described above. The study was approved by the local Ethics Committee (file number 8041/2020/BO2).

### 4.6. Statistics

Experimental results were processed by a spreadsheet program (Excel) and imported to SPSS statistics software (SPSS Inc. IBM, Armonk, NY, USA) for further analyses. We tested for normal distributions using the Shapiro–Wilk test. Differences in mean values were computed using either a Mann–Whitney test or a double-sided Student’s *t*-test with independent variables. Correlation analyses were computed by Spearman’s rank-order correlation. Correlations were displayed as values from −1 (linear negative correlation) to +1 (linear positive correlation) for ρ (rho). Differences in data sets with *p*-values smaller than 0.05 were considered statistically significant and indicated in the artwork where aprorpiate (*, **).

## 5. Conclusions

We conclude that the individual UM-UC lines seem to represent a cohort of bladder cancer cell clones with different; quite variable CD276 expression levels. The same applies—at least on transcript levels–to the expression of the stem cell markers ALDH1; CD24; and CD44 as well. Our recent studies provided evidence that elevated CD276 expression is recorded in all stages of bladder cancer [9]. We corroborate that the median CD276 expression on the UM-UC cell surfaces was higher when compared to NUCs, but elevated CD276 expression did not correlate with fast cell proliferation in vitro, and nor did the expression of the cancer stem cell markers investigated. We hypothesize that pathways regulating CD276—a bladder cancer cell and/or stem cell marker—are distinct and regulated independently from the other cancer stem cell markers investigated—i.e., CD24, CD44, and the ALDH1 paralog A1—and independent from the regulation of cell proliferation. However, the immune checkpoint molecule CD276 seems a rather dependable tool complementing cancer research when investigating both samples from bladder cancer tissue versus metastases. However, this conclusion must be viewed with caution as the cohort size of the clinical samples investigated in this study was low and only cells of the UM-UC line were explored in vitro in comparison to NUCs.

## Figures and Tables

**Figure 1 ijms-23-04969-f001:**
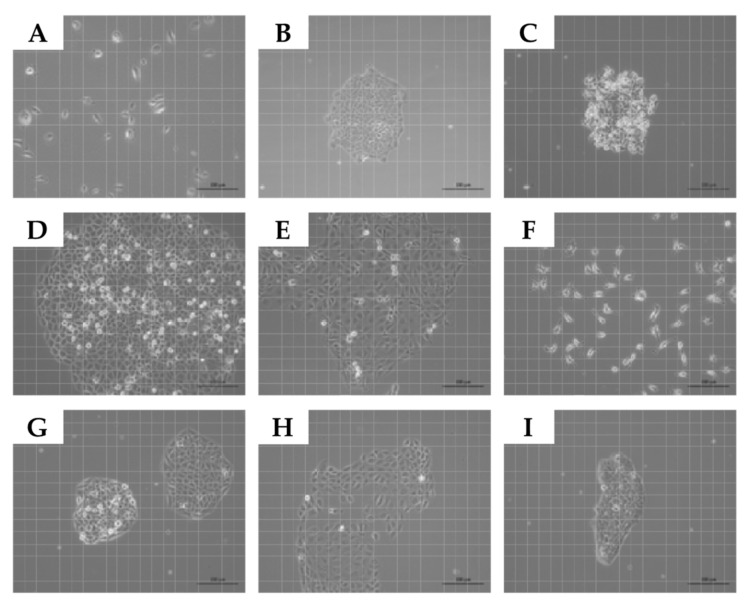
Growth patterns of urothelial cells. Normal urothelial cells (NUCs; (**A**)) and urothelial carcinoma cells (UM-UCs; (**B**–**I**)) were seeded in cell culture flasks and growth patterns were recorded by microscopy. NUCs and UM-UC-13 proliferated as individual cells and generated confluent layers in time. The other UM-UC lines grew in clusters. Size bars indicate 100 μM.

**Figure 2 ijms-23-04969-f002:**
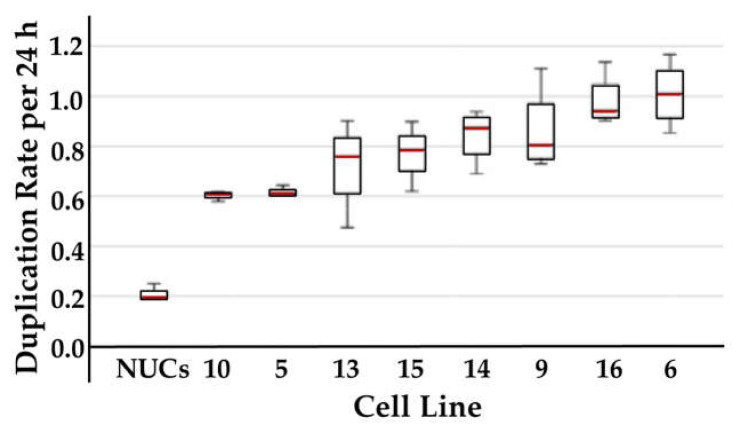
Cell proliferation rates of urothelial cells. Normal urothelial cells (NUCs) and urothelial carcinoma cells (UM-UCs) were seeded in cell culture flasks and duplication rates were counted over 3 consecutive passages. The red lines denote the mean proliferation rates.

**Figure 3 ijms-23-04969-f003:**
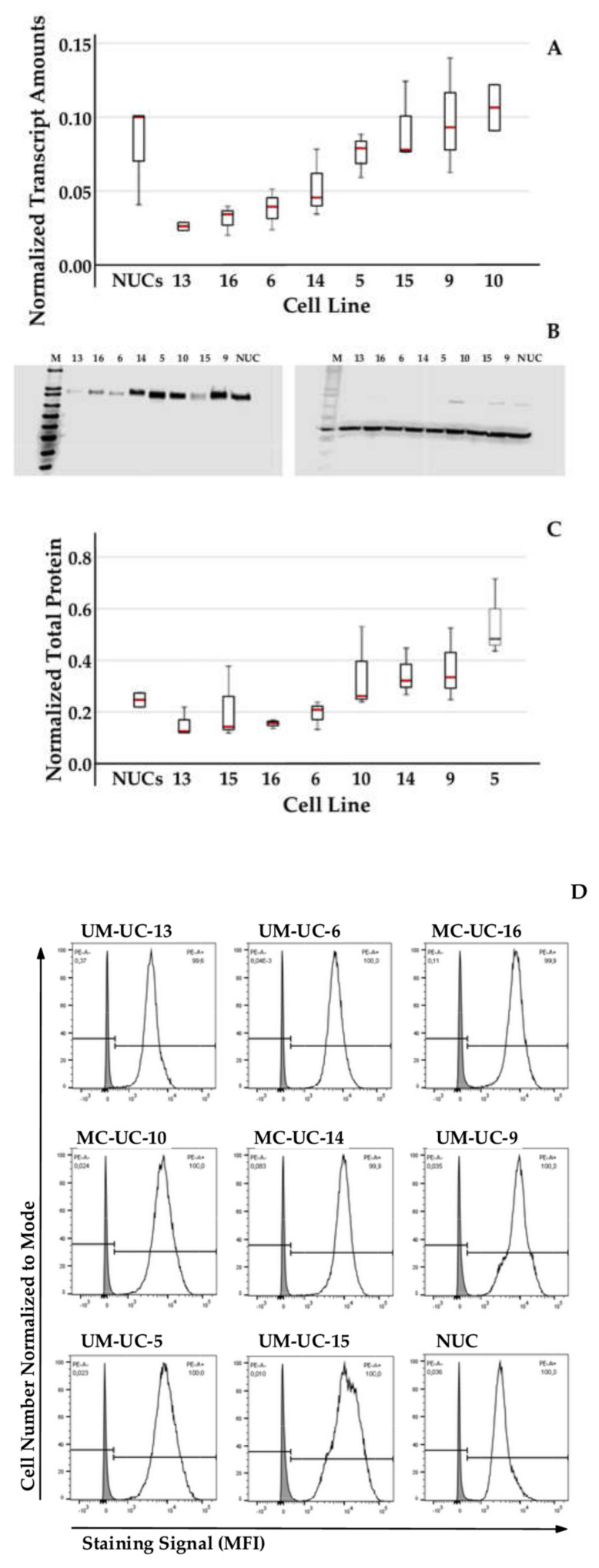
Expression of CD276 in urothelial cells. Expression of CD276 was investigated on the transcript (**A**), total protein (**B**,**C**), and cell surface (**D**–**F**) levels of NUCs and UM-UCs as indicated. (**A**) Steady-state mRNA expression was normalized to transcripts encoding 2 housekeeping controls. Red lines indicate the mean expression ± standard deviation from 3 individual analyses of cells as indicated. (**B**) Protein expression of CD276 was quantified by Western blot (left panel) and compared to expression of ß-actin (right panel). (**C**) Protein expression of CD276 was quantified by a blot scanner and normalized to the expression of ß-actin. Red lines indicate the normalized mean expression ± standard deviations. (**D**) The mean fluorescence intensity (MFI) of anti-CD276 staining of cells was investigated by flow cytometry of cells as indicated. The figure presents a representative experiment. The x-axis shows the log of the signal intensities, the y-axis the normalized cell numbers analyzed. Solid histograms denote CD276 staining, gray histograms the controls. (**E**) Comparison of the mean fluorescence intensities of CD276 staining of urothelial cells as indicated. (**F**) Mean fluorescence intensities of NUCs in comparison to all UM-UC cells investigated. Red lines indicate the mean expression lev.

**Figure 4 ijms-23-04969-f004:**
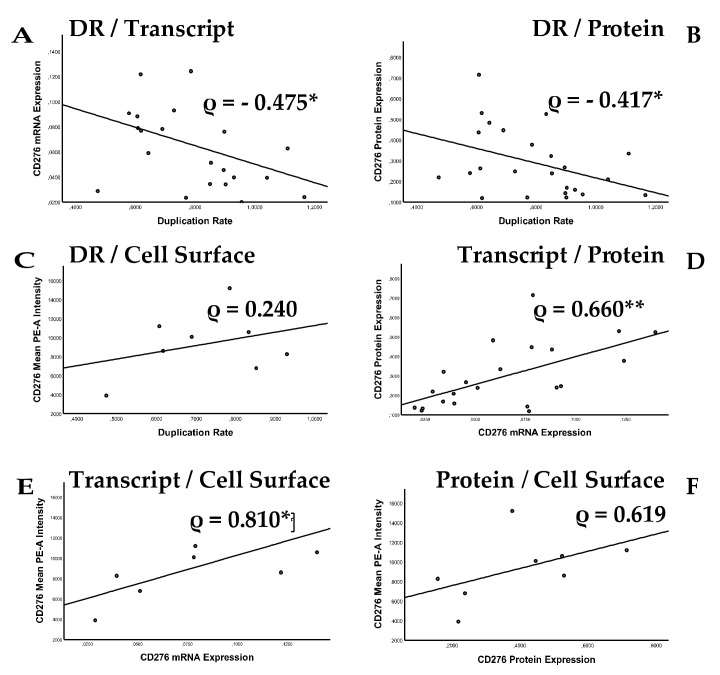
Correlation of CD276 expression with cell proliferation. Expression of CD276 was correlated on (**A**) transcript levels with cell duplication rates, (**B**) protein levels with duplication rates, (**C**) cell surfaces with duplication rates, (**D**) transcript versus protein levels, (**E**) transcript versus cell surface levels, and (**F**) protein versus cell surface levels. While the expression of CD276 transcript and protein yielded a negative correlation with proliferation, all other correlations were positive. Statistical significance is indicated (*, **).

**Figure 5 ijms-23-04969-f005:**
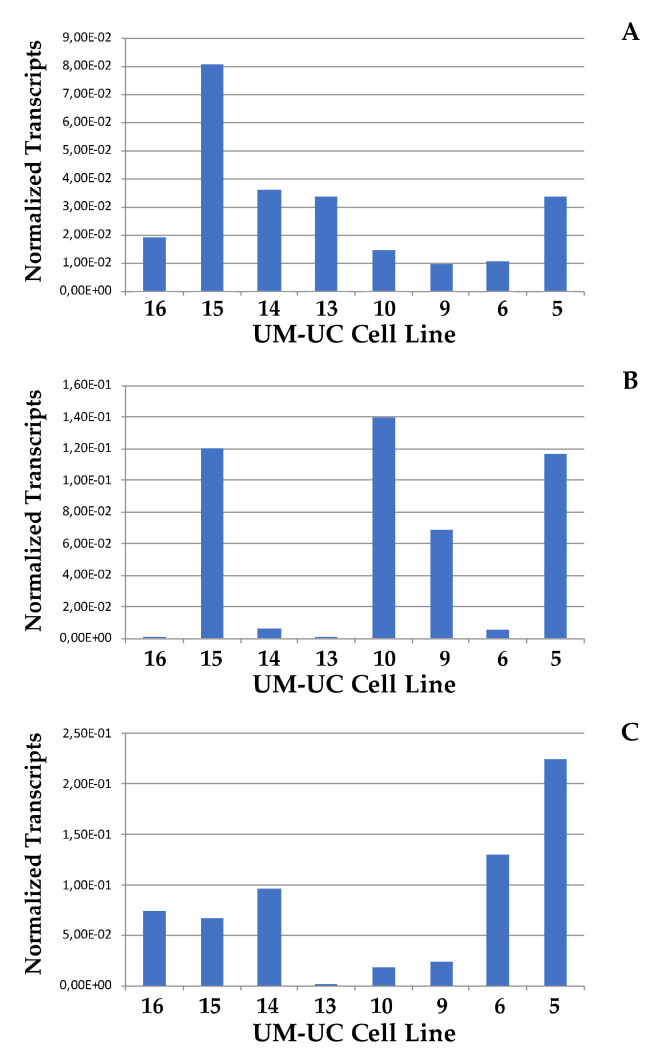
Expression of bladder cancer stem cell markers on UM-UCs. Expression of stem cell marker (**A**) ALDH1-A1, (**B**) CD24, and (**C**) CD44 was explored on transcript levels by RT-qPCR in UM-UC cell lines as indicated. The data document a representative analysis of target gene expression normalized to the respective housekeeping standards.

**Figure 6 ijms-23-04969-f006:**
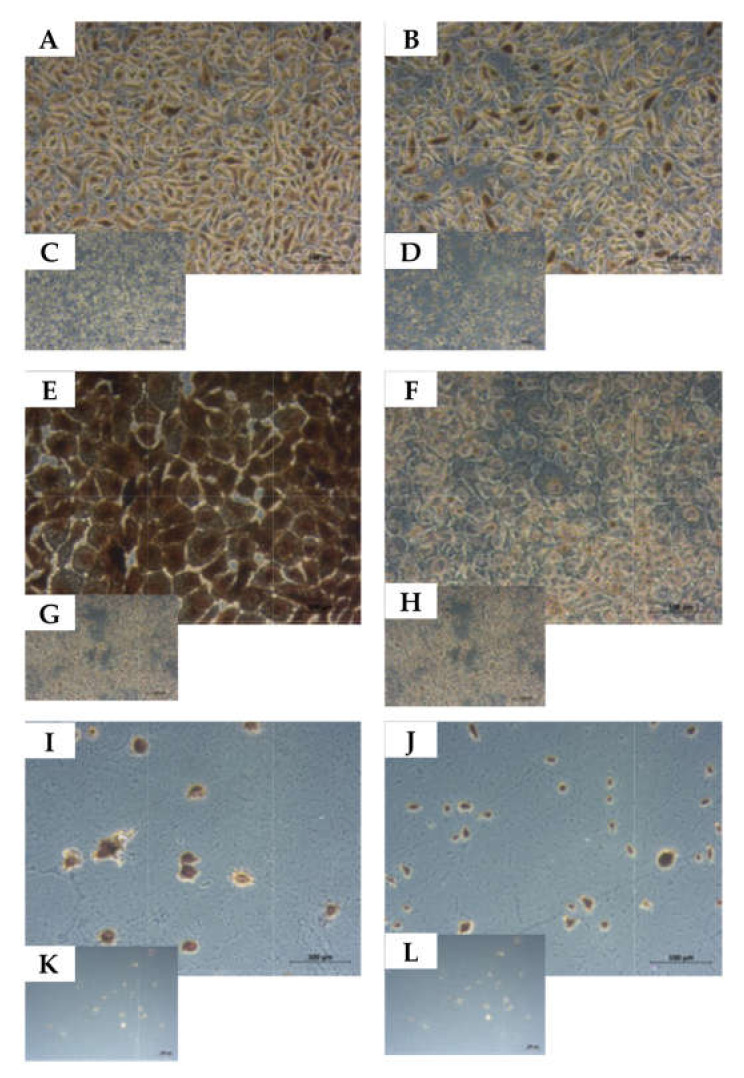
Detection of urothelial versus mesenchymal antigens on normal urothelial cells and UM-UC-10. Normal urothelial cells (**A**–**D**), UM-UC-10 (**E**–**H**), and UM-UC-13 (**I**–**L**) were stained with AE1/AE3 to detect cytokeratins as markers for urothelial cells (**A**,**E**,**I**) or with anti-vimentin antibodies to detect this mesenchymal lineage marker (**B**–**J**). Omitting the primary antibodies served as controls, respectively (**C**,**D**,**G**,**H**,**K**,**L**). Primary culture NUCs contained both, AE1/AE3^pos^ urothelial cells as well as vimentin^pos^ mesenchymal cells, while UM-UC-10 yielded a clear urothelial AE1/AE3^pos^ staining, but no vimentin staining above controls. UM-UC-13 detached upon fixation and staining and could therefore not be analyzed by immunocytochemistry (compare Figure 7). Size bars indicate 100 μM.

**Figure 7 ijms-23-04969-f007:**
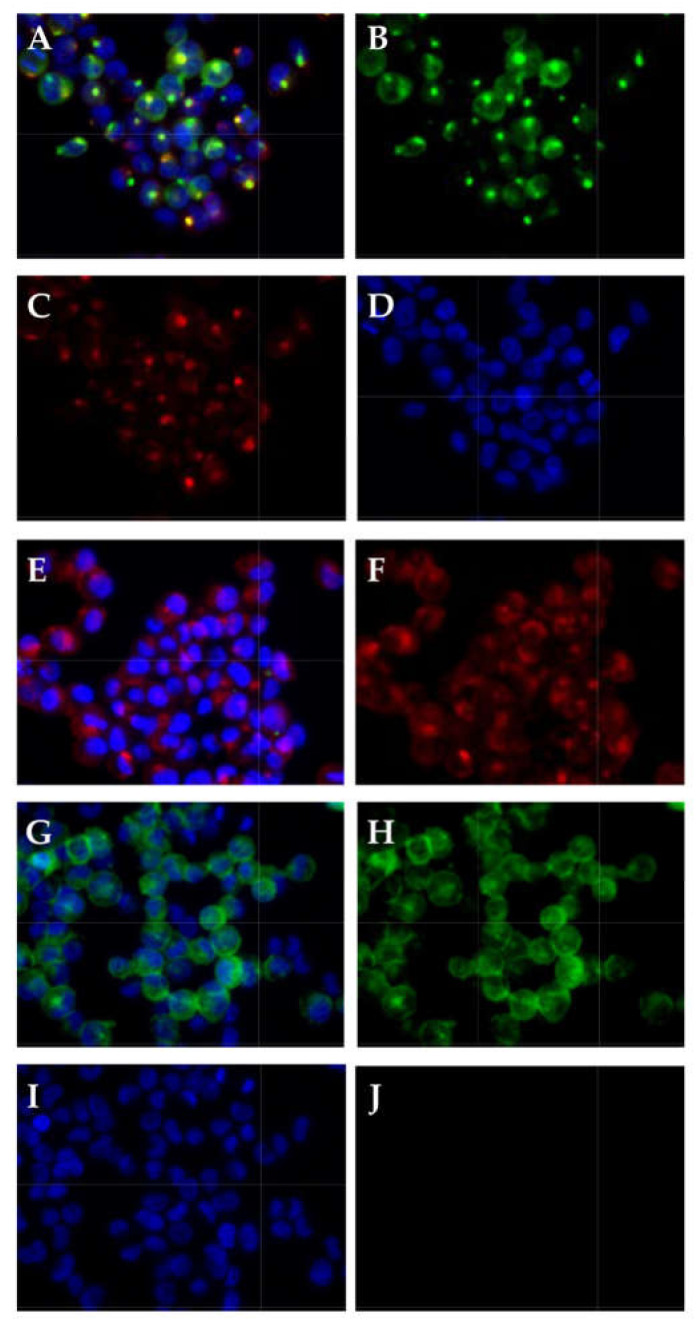
Detection of urothelial and mesenchymal lineage markers on UM-UC-13. (**A**) presents the merge of UM-UC-13 cells stained with antibodies reactive with (**B**) cytokeratin 5, (**C**) cytokeratin 8/18 and (**D**) counterstained by DAPI. (**E**,**F**) show staining of antibodies AE1/AE3 with or without DAPI. (**G**,**H**) show staining with anti-CD276 with or without DAPI, and (**I**,**J**) staining with anti-vimentin with or without DAPI.

**Figure 8 ijms-23-04969-f008:**
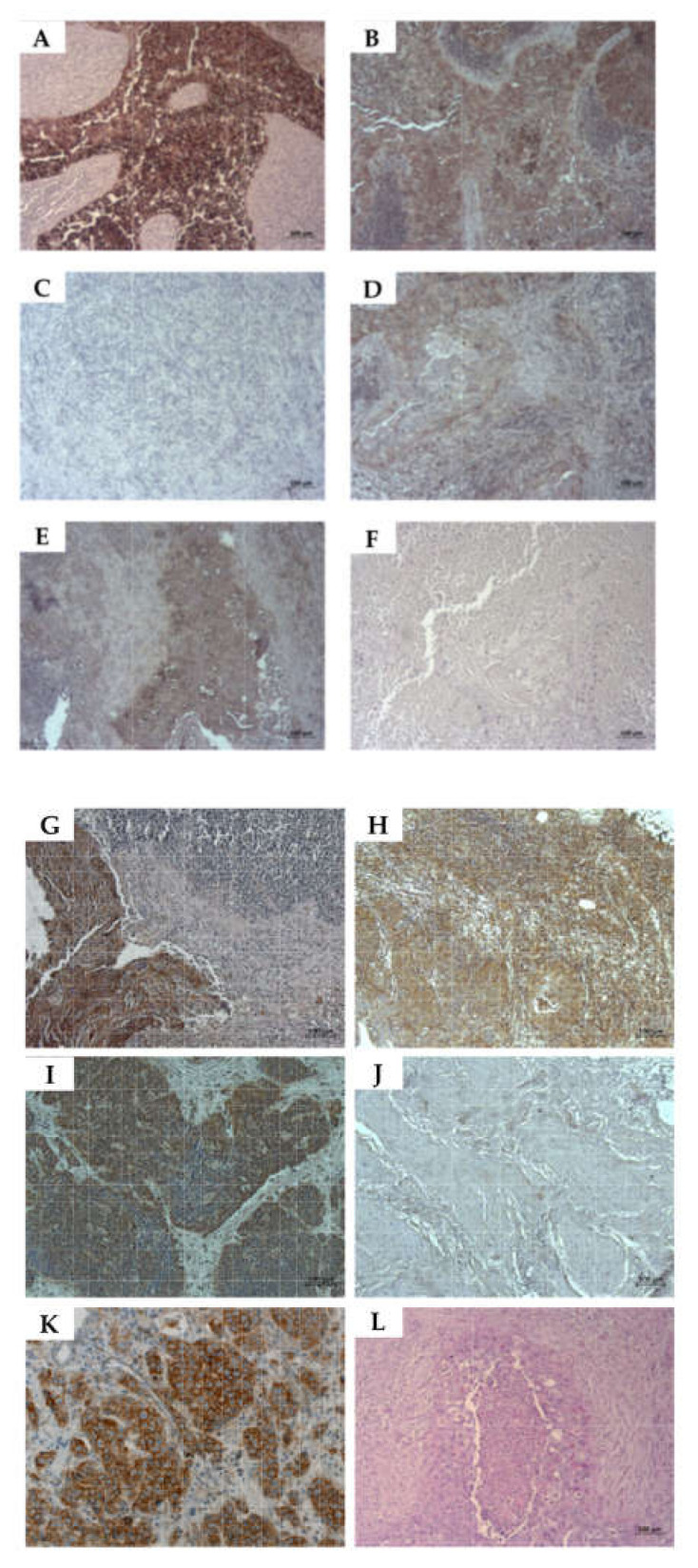
Expression of bladder cancer stem cell markers in tissue samples from bladder cancer patients. Paraffin sections were generated from metastases in lymph nodes (**A**–**F**) and from bladder tissue samples (**G**–**L**) of bladder cancer patients, and stained with antibodies AE1/AE3 (**A**,**G**), anti-vimentin (**B**,**H**), anti-CD24 (**C**,**I**), anti-CD44 (**D**,**J**), and anti CD276 antibodies (**E**,**K**), respectively. Omitting the primary antibodies served as controls (**F**,**L**). The samples were counterstained by HE. Expression of AE1/AE3, vimentin, and CD276 was recorded in both lymph node metastases and bladder cancer tissue sections. CD24 was not detected in any lymph node sample, but in bladder tissue. CD44 positive cells were seen in each lymph node sample investigated, while in the bladder cancer samples only a few cells were positive. Size bars indicate 100 μM.

**Figure 9 ijms-23-04969-f009:**
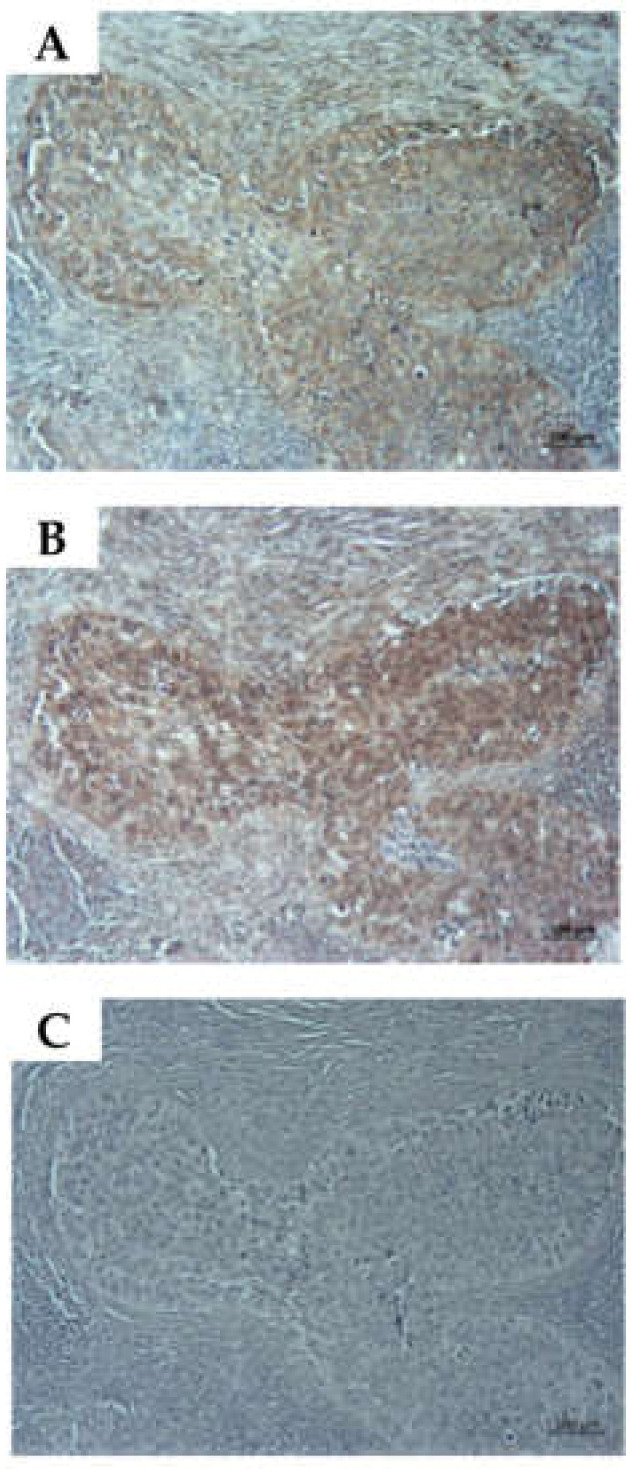
Expression of CD276 and proliferation marker Ki67 in metastases of bladder cancer patients. Paraffin sections from metastases in lymph nodes from bladder cancer patients were stained with (**A**) anti-CD276 and (**B**) with anti-Ki67 antibodies. (**C**) Infiltrating tumor tissue expressed both antigens at elevated levels. Size bars indicate 100 μM.

## Data Availability

Not applicable.

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
