# Peer review of "Elevated Expression of the Immune Checkpoint Ligand CD276 (B7-H3) in Urothelial Carcinoma Cell Lines Correlates Negatively with the Cell Proliferation"

_ijms, 2022, doi:10.3390/ijms23094969_

Round 1

Reviewer 1 Report

This article presented by Harland et al., for me personally, i will give high credits to their finding and the novelty of their study. However, i find several limits to this article. Although mostly should be blamed to its sample size, i think it should be corrected before i further get into review process.  

  1. First, there are several obvious type-2 error can be observed. For example, CD276 mean PE-A intensity is significantly correlated to CD276 mRNA expression, and CD276 mRNA expression is significantly correlated to duplication rate. In this way,  CD276 mean PE-A intensity should be significant related to the duplication rate or at least at alike trend as CD276 mRNA expression/duplication rate. However, CD276 mean PE-A intensity/duplication rate did not exhibit either. Please find out this kind of error in this article and state the possible error source in the discussion.
  2. I found that some of the results were computed by t-test. Please be noted that this parametric statistical method could be applied only after checking the skewness and kurtosis. In my opinion, the data in this article should less likely be applied with the parametric method. Please provide the skewness and kurtosis of the data. Otherwise, please re-calculate them. The wrong results might be lead to the wrong conclusions. I would like to suggest the author amend the related portions in the article, and any conclusions made based on it previously. After the adjustment, the rest portion could be advanced to review process again. 

Author Response

Critique 1: First, there are several obvious type-2 error can be observed. For example, CD276 mean PE-A intensity is significantly correlated to CD276 mRNA expression, and CD276 mRNA expression is significantly correlated to duplication rate. In this way, CD276 mean PE-A intensity should be significant related to the duplication rate or at least at alike trend as CD276 mRNA expression/duplication rate. However, CD276 mean PE-A intensity/duplication rate did not exhibit either. Please find out this kind of error in this article and state the possible error source in the discussion.

Ad Reviewer 1: We appreciate this comment and revised the manuscript accordingly. Chapter 2, lines 164 – 165 and the paragraph below were rephrased to comply with the critiques. In chapter 3, lines 349 – 363 and lines 454 – 456 were rephrased to discuss the complexity of regulation of CD276 in more detail. The numbers of lines mentioned correspond to the PDF document “Changes-man-...” which includes all revisions made.

Reviewer 2 Report

The publication by Harland et al. describes the study of CD276 molecule expression in urothelial carcinoma cell lines and in patient cancer samples and in metastases in lymph nodes. The authors examined whether CD276 expression correlated with cell proliferation, expression of molecular markers of UC by methods of immunohistochemistry and fluorescent microscopy. The level of CD273 expression in all cell lines was confirmed on level of expression analysis, protein detection, cell surface detection. Negative correlation of duplication rate of cell lines with CD276 expression was shown to be statistically significant. The expression of CD276 did not correlate with expression of UC stem cell markers. The last part of investigation was dedicated to analysis of tissues of patients primary cancer and metastases (Fig. 8). The data obtained can be characterized as heterogenous.

Critical points:

Introduction provides the survey of knowledge about urothelial cancer and CD276. However, experimental part contains measurements of expression of characteristic markers of urothelial cancer cells or mesenchymal cells such as ALDH-A1, CD24, CD44, cytokeratins and vimentin. Short survey about connection of these markers with urothelial cancer cells and their meaning for diagnostics should be added to Introduction. For better orientation in cell lines, I recommend to add a Table of known characteristics, origin, number of passages etc. of each cell line.

Minor points:

The criticism of the manuscript concerns rather the data presentation than the results themself.

Fig.1 -Each picture must be denoted by the name of the cell line in the legend.

Fig. 3 and 8 - Marks of pictures (A,B,C…) are in inverted order The flow of pictures must be uniform through entire text. It should start with A on left side and continue to the right side.

Fig. 2, 3- I recommend the same order of cell lines in all graphs. It would be better for comparison.

Fig. 5- Show data on all three markers in one graph only. The bars for three molecules mark with different colours.

Fig. 6- pictures I -L should be omitted as they do not provide any data.

Fig. 8, 9- It would be helpful to insert numbers of positive/tested samples for each marker in every picture.

Fig. 9- The origin of information about phenotype of line UM-UC-13 (row 212) is not obvious, as Figs. 2, 3 show not Ki67 expression.

Author Response

I found that some of the results were computed by t-test. Please be noted that this parametric statistical method could be applied only after checking the skewness and kurtosis. In my opinion, the data in this article should less likely be applied with the parametric method. Please provide the skewness and kurtosis of the data. Otherwise, please re-calculate them. The wrong results might be lead to the wrong conclusions. I would like to suggest the author amend the related portions in the article, and any conclusions made based on it previously. After the adjustment, the rest portion could be advanced to review process again.

Ad Reviewer 2: We thank the reviewer for the thorough analysis of the statistical methods. We clarified in Material and Methods, that we checked for normal distribution using Shapiro-Wilk test before applying the t-test and the respective p-value of .374 confirming this. We further added skewness (.29) and kurtosis (-0.866) to the results. The changes can be seen in chapter 2, lines 145 – 147, and in chapter 4, lines 614 -618. Again, the numbers of lines mentioned correspond to the PDF document “Changes-man-...” which includes all revisions made.

Round 2

Reviewer 2 Report

Manuscript was modified with respect to criticism of both reviewers. I recommend accepting it.